# The Potential of Novel Lipid Agents for the Treatment of Chemotherapy-Resistant Human Epithelial Ovarian Cancer

**DOI:** 10.3390/cancers14143318

**Published:** 2022-07-07

**Authors:** Mark W. Nachtigal, Alon D. Altman, Rajat Arora, Frank Schweizer, Gilbert Arthur

**Affiliations:** 1Department of Biochemistry and Medical Genetics, University of Manitoba, Winnipeg, MB R3E 0J9, Canada; gilbert.arthur@umanitoba.ca; 2Department of Obstetrics, Gynecology and Reproductive Sciences, University of Manitoba, Winnipeg, MB R3E 0J9, Canada; alon.altman@cancercare.mb.ca; 3CancerCare Manitoba Research Institute, CancerCare Manitoba, Winnipeg, MB R3E 0V9, Canada; 4Department of Chemistry, University of Manitoba, Winnipeg, MB R3E 0J9, Canada; arorar5@myumanitoba.ca (R.A.); frank.schweizer@umanitoba.ca (F.S.)

**Keywords:** epithelial ovarian cancer (EOC), chemotherapy resistance, methuosis, glycosylated antitumor ether lipid

## Abstract

**Simple Summary:**

Disease recurrence and chemotherapy resistance are the major causes of mortality for the majority of epithelial ovarian cancer (EOC) patients. Standard of care relies on cytotoxic drugs that induce a form of cell death called apoptosis. EOC cells can evolve to resist apoptosis. We developed drugs called glycosylated antitumor ether lipids (GAELs) that kill EOC cells by a mechanism that does not involve apoptosis. GAELs most likely induce cell death through a process called methuosis. Importantly, we showed that GAELs are effective at killing chemotherapy-resistant EOC cells in vitro and in vivo. Our work shows that the EOC community should begin to investigate methuosis-inducing agents as a novel therapeutic platform to treat chemotherapy-resistant EOC.

**Abstract:**

Recurrent epithelial ovarian cancer (EOC) coincident with chemotherapy resistance remains the main contributor to patient mortality. There is an ongoing investigation to enhance patient progression-free and overall survival with novel chemotherapeutic delivery, such as the utilization of antiangiogenic medications, PARP inhibitors, or immune modulators. Our preclinical studies highlight a novel tool to combat chemotherapy-resistant human EOC. Glycosylated antitumor ether lipids (GAELs) are synthetic glycerolipids capable of killing established human epithelial cell lines from a wide variety of human cancers, including EOC cell lines representative of different EOC histotypes. Importantly, GAELs kill high-grade serous ovarian cancer (HGSOC) cells isolated from the ascites of chemotherapy-sensitive and chemotherapy-resistant patients grown as monolayers of spheroid cultures. In addition, GAELs were well tolerated by experimental animals (mice) and were capable of reducing tumor burden and blocking ascites formation in an OVCAR-3 xenograft model. Overall, GAELs show great promise as adjuvant therapy for EOC patients with or without chemotherapy resistance.

## 1. Introduction

Epithelial ovarian cancer (EOC) continues to be a lethal malignancy in women worldwide, and resistance to chemotherapy remains one of the main contributors to EOC patient morbidity and mortality [1,2]. The early disease has a relatively good prognosis, and this fact led to screening and symptomatic-based studies [3,4,5,6,7] in the hopes of detecting disease earlier and improving outcomes. Unfortunately, none of these studies have proven useful for increasing survival. Other studies examined the role of time-to-diagnosis and time-to-care and once again found no significant benefit for survival [8,9]. Improved surgical techniques resulting in optimal tumor debulking, improved cytotoxic delivery methods, and novel non-cytotoxic chemotherapeutics are treatment modalities that have contributed to improved progression-free and overall survival in the EOC patient population in the past several decades. Herein, we review the development and utility of a novel glycolipid compound belonging to the glycosylated antitumor ether lipid (GAEL) class of drugs for the potential treatment of EOCs.

Historically, EOC was treated with multiple different agents until the late 1990s, when several studies confirmed the superiority of cisplatin plus paclitaxel as the basis of EOC chemotherapeutic treatments [10,11]. Subsequent trials showed the equivalence of carboplatin to cisplatin, with decreased toxicity, leading to the fundamental base of modern cytotoxic therapy for EOC [12,13]. While the majority of high-grade serous ovarian cancer (HGSOC) patients respond to initial chemotherapy (6–9 cycles of a platinum agent (carboplatin) and a taxane (paclitaxel)), up to 75% of EOC patients will relapse within 18 months with chemotherapy-resistant disease [14,15]. Moreover, mucinous, clear cell and low-grade serous EOC histotypes are characterized by a poor response to chemotherapy. EOC therapy has continued to develop with changes in surgical approaches [16,17,18], different approaches for dosing of chemotherapy (e.g., dose dense and dose intense) [19,20], chemotherapy delivery (e.g., intraperitoneal and hyperthermic intraperitoneal) [21,22], additional adjuvant agents (e.g., Bevacizumab) [23,24,25,26], and maintenance therapy [27,28]. The most promising recent advances in EOC management have surrounded advancements in poly-ADP ribose (PARP) inhibitor therapies, specifically for *BRCA1*/*2* and homologous recombination-deficient (HRD) patients [29,30,31,32,33,34]. Ongoing trials with immune modulators have not yet shown benefits for this patient population [35]. Despite great advances in care, overall survival, although improved slightly, has not yet made drastic changes in several decades [36,37,38,39,40,41]. Thus, additional treatments capable of preventing tumor growth and effectively killing chemoresistant EOC cells are desirable. The connection between translational research and clinical work will be critical in the advancement of gynecologic oncology and cancer therapy.

As part of our ongoing search for novel treatments for human epithelial cancers, we developed glycosylated antitumor ether lipids (GAELs; Figure 1) and demonstrated that GAELs exhibit anticancer activities in a diverse group of human cancer types in vitro. GAEL compounds are structurally related to antitumor ether lipids (AELs), including edelfosine [42], miltefosine [43], erucylphosphocholine [44] and perifosine [45] that have been tested as anticancer agents in clinical studies. GAELs would fall under the class of cytotoxic rather than targeted agents. GAELs are able to kill chemosensitive and chemoresistant human high-grade serous ovarian cancer (HGSOC) cell lines and patient samples grown as adherent or 3D spheroid cultures [46,47,48,49,50,51,52]. Importantly, the GAEL mechanism of action is apoptosis-independent. This characteristic distinguishes GAELs from most of the currently used anticancer drugs that rely on apoptosis to kill tumor cells. Because cancer cells often overcome apoptosis induced by many cytotoxic agents, the addition of GAELs would be an excellent complementary treatment to overcome drug-resistance mechanisms. Moreover, as cellular genetic heterogeneity is observed in EOCs, the ability of GAELs to kill chemotherapy-sensitive and resistant cells would be an additional benefit of including GAELs as adjuvant treatment. This review introduces how GAELs were developed from observations of the biological activity of lysophosphatidylcholine (LPC), provides a brief overview of structure–activity relationship (SAR) studies, and highlights our studies testing GAEL activity to kill chemotherapy-sensitive and -resistant human EOC cells and the likely mechanism of action contributing to GAEL-induced cell death, namely methuosis. We postulate that induction of methuosis in chemotherapy-resistant EOC cells identifies a novel cell death pathway for the further development of new treatments for EOC patients.

## 2. GAELs: How Did We Get Here?

Antitumor ether lipids (AELs) are broadly defined as a group of synthetic ether lipid analogs with antitumor activity. They are comprised of three subclasses including alkyllysophospholipids (ALPs), alkylphospholipids (APLs), and GAELs (Figure 1). The discovery of the antitumor activity of ether lipids originated from studies at the Max-Planck-Institut für Immunobiologie in Freiburg, Germany, in the 1960s. LPC, which is generated from hydrolysis of phosphatidylcholine by phospholipase A2, was found to strongly potentiate the phagocytic activity of peritoneal macrophages in vitro and in vivo [53,54,55]. LPC is not a viable drug because it is rapidly metabolized and inactivated in cells by acyltransferases and lysophospholipases. To overcome this, metabolically stable LPC analogs were synthesized by replacing the acyl group at the sn-1 position with an alkyl group with 12–20 carbons. The sn-2 position was either an OH, H, or a methoxy (OCH_3_) group [56,57,58]. These ether LPC analogs, the ALPs (Figure 1), demonstrated increased half-life and significantly higher ability to boost antibody production in response to various antigens compared to LPC [59]. Thus, ALPs were initially developed for use as potential immunomodulators. Subsequently, a number of other ether LPC analogs were observed to have cytotoxic activity against a variety of cancer cells in vitro with 1-O-octadecyl-2-O-methyl-glycerophosphocholine (ET-18-OCH_3_; aka edelfosine) being the most active [60,61,62]. Further testing of ALPs using in vivo mouse and rat tumor models demonstrated antitumor activity by reducing tumor growth and metastasis [63,64]. While several different pathways contributing to cell death in vitro have been examined, including inhibition of protein kinase C [65,66], decreasing arachidonate release [66], abrogating phosphatidylcholine biosynthesis [67,68], and altering cell membranes [69,70], edelfosine exhibited antitumor activity against a wide range of cancer cells by inducing apoptosis [71,72]. ALPs have been tested in clinical trials and exhibit high tolerance to reversible toxicities and produce tumor stasis or a decrease in the tumor growth rate [63,64,71,73,74,75,76,77]. Despite having promising clinical efficacy in vivo, the compounds are not indicated for the treatment of human cancers.

Studies to identify the minimum structure of ALPs required for anticancer activity led to the development of the alkylphospholipids (APLs; Figure 1). These compounds lack the glycerol backbone and consist of an alkylchain esterified to a phosphobase. The prototype of this group is hexadecylphosphocholine (miltefosine) [71,78]. While miltefosine was initially investigated as an anticancer agent, tests were stopped due to toxic side effects. It is currently used for leishmaniasis and amoeboid infections. Several analogs of APLs, including erucylphosphocholine [79] and perifosine [80,81], have been synthesized and their antitumor activities characterized [82]. Perifosine was under development to treat human cancers as it blocks PI3K/AKT/mTOR signaling [83]. Perifosine showed considerable promise in early phase clinical trials but failed in Phase III trials for colorectal cancer and refractory multiple myeloma [84]. The mode of action of these compounds appears to be similar to that of the ALPs in killing cells by inducing apoptosis [79,82].

GAELs were developed by replacing the phosphocholine of edelfosine with a monosaccharide moiety via an O-glycosidic bond [85]. The cancer cell killing activity of the GAEL compounds that were initially synthesized was weak compared to that of edelfosine. Substitution of the O-glycosidic to an S-glycosidic bond to improve lipophilicity did not significantly change the cancer cell-killing characteristics of the GAEL compounds [86,87]. Subsequent structure–activity relationship (SAR) studies determined that replacing the OH at position 2 of the glucose with an NH_2_ yielded cancer cell-killing activities that were similar or superior to edelfosine [88]. This discovery opened up the field and led to the development of potent analogs and the further identification and characterization that GAELs were distinct from ALPs and APLs by inducing a non-apoptotic (caspase-independent) form of cell death.

## 3. GAEL Structure–Activity Relationship

Several GAEL prototypes have been described [50,52,88,89,90,91,92,93] (Figure 2). The key event to the development of these compounds was the discovery by the Bittman group that the replacement of neutral thio-glucose by a basic 2-amino-β-D-glucose scaffold produced a potent GAEL [β-GLN (1-O-hexadecyl-2-O-methyl-3-O-(2′-amino-2′-deoxy-β-D-glucopyranosyl)-sn-glycerol)] with antiproliferative effects (Figure 2, compound **2**). Cell killing activity was tested in various human epithelial cancer cell lines, including A549, MCF-7, A427, and T84, with the greatest sensitivity in HGSOC OVCAR-3 cells [88]. Indeed, compared to edelfosine, β-GLN was five times more effective at killing OVCAR-3 cells [88]. Further experiments showed that acylation of the basic 2-amino group in β-GLN resulted in a 2–3-fold reduction in anticancer activity, indicating that the basic primary 2-amino group is critical for the antitumor effect. Further modification of β-GLN to contain an α-glucosidic linkage of 2-amino-D-glucose (α-GLN; Figure 2, compound **1**) displayed two-fold enhanced cell-killing activity compared to β-GLN [91]. By contrast, replacement of the O-glycosidic linkage with S-glycosidic linkage and modifications on the glycero-ether moiety, in addition to guanidinylation of the amino group, resulted in reduced cell-killing effects [91]. In order to increase the metabolic stability of β-GLN, C-glycosidic analogs were generated (Figure 2, compounds **3** and **4**) that retained antiproliferative effects [92,93]. Studies to replace the α-O-glucosidic linkage with an α-O-galactosidic or α-O-mannosidic linkage revealed that both α-O-galactosidic and α-O-glucosidic linkages displayed comparable cell-killing effects while a >4-fold reduction in cell killing activity was observed with the α-O-mannosylated analog [52]. Introducing a second amino group at the 6-position in α-GLN or β-GLN (Figure 2, compounds **5** and **6**) enhanced the cell-killing effect. Interestingly, both dibasic analogs displayed potent anticancer stem cell (CSC) activity against BT-474 and DU-145 stem cells with superior activity compared to salinomycin [50]. Dibasic analogs of α-GLN or β-GLN were produced in which *D*-glucose (*D*-Glc) was replaced by glycosidase-resistant *L*-glucose (*L*-Glc) (Figure 2, compounds **7** and **8**) [90]. Moreover, decoration of the dibasic *L*-Glc scaffold by attachment of a phenyl group further enhanced the cell-killing effect [90]. The incorporation of a third amino group and amphiphilic modulation of the *D*-Glc scaffold also enhanced the cell-killing effects of GAELs [94]. While first-generation GAELs were synthesized with *D*-linked-sugars, their potential susceptibility to circulating and cellular glycosidases in vivo led to the development of *L*-sugar-linked GAEL analogs that demonstrated equivalent or enhanced activity [90]. Humans do not metabolize *L*-sugars; thus, *L*-GAELs are predicted to have longer half-lives in vivo. A structure–activity map that summarizes the structural modifications of the antitumor effect is shown in Figure 3.

One of the challenges to exploiting GAELs for therapeutic applications and animal studies is the requirement for multi-step synthesis resulting in low isolation yields. Synthetic processes required 12 or more synthetic steps. To reduce the number of synthetic steps, a more easily accessible *L*-rhamnose-based cationic lipid was generated (Figure 2, compound **9**; 3-amino-1-*O*-hexadecyloxy-2R-(*O–*α-*L*-rhamnopyranosyl)-*sn*-glycerol (*L*-Rham)) [95]. This *L*-Rham-based GAEL differs from previous GAELs by the glycosidic linkage and glycero backbone; this compound retained cell-killing effects and was found to possess antitumor effects and therefore was used for further in vitro and in vivo studies [49,95]. GAELs possess many desirable anticancer characteristics that include: (1) killing chemoresistant cell lines and primary HGSOC cells grown as 2D or 3D cultures and causing the disaggregation of 3D spheroids [83,84,92,93]; (2) killing human cancer stem cell-enriched fractions from breast and prostate cell lines [83,84,94]; and (3) killing cells via an apoptosis-independent mechanism of action [47,48,50,51,52]. This latter characteristic distinguishes GAELs from most of the currently used chemotherapeutic drugs that rely on apoptosis to kill HGSOC cells.

## 4. GAEL Cell-Killing Activity against Epithelial Ovarian Cancer

Intriguingly, the sensitivity of the NIH OVCAR-3 HGSOC cell line to β-GLN compared to other AELs stood out in our research testing the effect of GAELs on human epithelial cancer cell viability [88]. Therefore, the sensitivity of EOC cells was further investigated using drug-resistant EOC cell lines as well as patient samples obtained from chemosensitive and chemoresistant patients [48]. The cell-killing effect of β-GLN and compound **5** (1-O-Hexadecyl-2-O-methyl-3-O-(2′,6′-diamino-2′,6′-dideoxy-α-D-glucopyranosyl)-sn-glycerol; Figure 2), were tested using the A2780-s (cisplatin sensitive) and A2780-cp (cisplatin-resistant) endometrioid EOC cell lines, the COV362 HGSOC cell line, as well as primary samples isolated from EOC patients [96,97] diagnosed with high grade serous and clear cell ovarian carcinoma. Culture systems utilized adherent monolayers and non-adherent 3-dimensional (3D) cultures. Cell viability was determined using MTS assays and flow cytometry labeling cells with Annexin V-APC and 7-amino actinomycin D [48]. To provide a relevant drug for comparison, A2780-cp and chemoresistant patient samples were treated with cisplatin. These cells showed a high tolerance for cisplatin treatment [inhibitory concentration resulting in 50% cell numbers (IC_50_) was 30 µM for A2780-cp, whereas ~50% of the cell population in primary cultures continued to grow even in 90 µM cisplatin]. In contrast to this cisplatin profile, EOC cells were sensitive to GAEL treatment, e.g., IC_50_ for compound **5** was A2780-cp = 9 µM, COV362 = 2.5 µM, and primary EOC samples demonstrated an IC_50_ ranging from 2.5–3.5 µM. Even greater sensitivity was observed at lower concentrations when 3D spheroids were treated for up to 72 h (IC_50_ ranging from 0.2 to 1 µM). Indeed, spheroid integrity was slowly eroded over time, leading to the disintegration of spheroids and cell death. Importantly, these results showed that GAEL compounds can effectively kill cisplatin-sensitive and cisplatin-resistant EOC cell lines and primary cells isolated from patient samples grown as adherent monolayers or non-adherent spheroids.

Our previous structure/activity studies aimed to improve the activity and metabolic stability of GAELs, which led to the development of *L*-GAELs [90,95]. While *D*-sugar-linked GAELs would be susceptible to circulating and cellular glycosidases in vivo, thus rendering them inactive, *L*-sugar-linked GAEL analogs cannot be metabolized in humans. We tested the cell-killing effect of *L*-Rham on chemonaïve, chemosensitive, and chemoresistant HGSOC cell lines and primary samples derived from HGSOC patient ascites [49]. In addition, these in vitro studies were complemented by testing *L*-Rham tolerability and efficacy in vivo using two in vivo models: (1) a COV362 xenograft in a chicken allantoic membrane (CAM) model and (2) an OVCAR3 xenograft murine model [49].

The in vitro sensitivity of HGSOC cell lines (CaOV3, COV362, OVCAR3, TOV1946) and primary HGSOC cell samples (N = 11) to *L*-Rham was tested (Figure 4). Our data showed an IC_50_ ranging from 4.8 to 13 µM when grown as 3D cultures. Importantly, we observed the disintegration of spheroid structures, similar to our observations with previous GAELs (Figure 5) [85,91,92]. Additionally, we observed a leftward shift in sensitivity when spheroids were exposed to *L*-Rham over time (up to 96 h). For example, the IC_50_ for patient sample EOC 180 changed from 13.4 µM at 48 h to 8.7 µM at 96 h [49]. Similar to our previous studies with β-GLN and compound **5** [48], these results demonstrated that treatment with *L*-Rham is effective at killing chemonaïve and chemoresistant HGSOC patient samples in vitro.

To complement the in vitro results and provide the first evidence showing GAEL efficacy in vivo, COV362 HGSOC cells were used to generate xenografts in a CAM model (N = 18 eggs per condition). Experiments with chick embryos have been recognized as an alternative to mouse xenografts for in vivo experiments by the National Center for the Replacement, Refinement, and Reduction of Animals in Research (NC3R, UK), and all experiments complied with European Directive 2010/63/EU. Fertilized White Leghorn eggs are incubated at 37.5 °C with 50% relative humidity for 9 days. At this time (E9), the CAM was accessed by drilling a small hole through the eggshell into the air sac, and a 1 cm^2^ window was cut in the eggshell above the CAM. An inoculum of 3 × 10^6^ COV362 cells was added onto the CAM of each egg. On day 10 (E10), COV362 tumors began to be detectable. *L*-Rham treatment was initiated two days after cell injection with doses approximately equal to 0.44–0.88 µM in ovo. Paclitaxel (0.11 µM in ovo) was used as a positive treatment control. Eggs were treated for a total of 8 days and harvested on day 18 prior to normal hatching on day 21. Our studies showed that *L*-Rham was as effective as paclitaxel in reducing COV362 tumor size and metastatic spread [49]. Moreover, no toxicity or macroscopic effect on the development of chick embryos was observed.

Dose range finding and maximum tolerated dose experiments were conducted by the BC Cancer Agency Investigational Drug program using 6–8-week-old female Rag2M and NRG immunocompromised mice in compliance with the Canadian Council on Animal Care guidelines. Drug delivery (intravenous or intraperitoneal) and drug vehicle (1% propylene glycol or 0.9% saline) studies were conducted [49]. It should be noted that phosphate-buffered saline should not be used to resuspend GAELs as this causes drug precipitation. We determined that the best dose and method of delivery was intraperitoneal injection using 0.9% saline as a drug vehicle. Mild clinical signs of toxicity were observed, but these resolved prior to necropsy. Drug treatment that was tested, but not tolerated, included multiple doses of *L*-Rham delivered i.p. at 40 mg/kg in 1% PG (MWFx2) that caused abdominal tenseness and distension, body weight loss, pain, coat piloerection, pale extremities, and dehydration. *L*-Rham showed tolerable toxicity at orders of magnitude higher than those that kill primary HGSOC cell spheroids in vitro and suggested that *L*-Rham may be well tolerated at therapeutic doses.

Further in vivo evidence for the antitumor effect of *L*-Rham was shown using OVCAR3 intraperitoneal xenografts in an endpoint study (74 days) [49]. The efficacy of *L*-Rham in reducing OVCAR3 tumor xenografts in immunodeficient NRG mice was assessed in low and high tumor burden models with early or late drug intervention (Table 1).

For the low tumor burden model, *L*-Rham treatment was initiated 7 days after cell injection, whereas the high tumor burden model started *L*-Rham treatment 40 days after cell injection. Mice received eight doses of *L*-Rham every four days (32 days of treatment). *L*-Rham effectively reduced tumor formation in four out of six mice in the low tumor burden group, while no effect was observed if animals had a high tumor burden (six out of six mice showed tumor burden equivalent to vehicle-treated animals). Importantly, *L*-Rham blocked ascites formation in 100% of low and high tumor burden animals. Our research suggests that *L*-Rham might best serve as an adjuvant agent following optimal surgical debulking and standard-of-care chemotherapy (carboplatin + paclitaxel) in order to produce a low tumor environment where *L*-Rham would effectively reduce tumor recurrence and block ascites formation.

## 5. GAEL Mechanism of Action

While the molecular mechanism of action contributing to GAEL-induced cell death is not completely known, research to investigate the GAEL mechanism of action for cell killing has resulted in the identification of several key characteristics. One of the first features identified is that GAELs kill cells via an apoptosis-independent mechanism [48,49,51]. We determined that GAELs can induce cell death in cells lacking *Casp3*, *Casp9*, or *Map3k5* [51] and cell death persists in the presence of pan-caspase inhibitors (z-VAD-FMK or Q-VD-OPh) [48,49,51]. This distinguishes GAELs from the standard cytotoxic agents used to treat EOC (e.g., carboplatin, paclitaxel, liposomal doxorubicin).

A unifying phenotypic response to GAEL treatment is the formation of phase-lucent cytoplasmic vesicles (Figure 6). GAEL-induced cell death is intimately related to the generation of lysosome-associated membrane protein 1 containing acidic cytoplasmic vacuoles [93,95,97,98], features found in endosomes and lysosomes. Initial interpretations suggested vesicle formation may result from autophagy. However, GAEL-induced cell death persists in *Atg5* null mouse embryonic fibroblast cells [47]. In addition, experimental evidence suggested that GAEL-induced cell death may involve altering lysosomal permeability to allow the release of acid proteases such as cathepsin D [47]. We previously reported that after cells were incubated with β-GLN, cathepsins B, D, and L were released into the cytosol, and an increase in cathepsin activity was detected [47]. Furthermore, in cells pretreated with pepstatin A, an inhibitor to cathepsin B, cell death was partially inhibited in response to β-GLN. A more limited study in EOC cells (encompassing HGSOC and clear cell histotypes) revealed that pepstatin A did partially attenuate GLN-induced cell death in some primary EOC cell samples but not in others [48]. Thus, this is not a universal response. We have also shown that alteration of mitochondrial membrane potential, which is essential for paraptosis or oncosis, two non-apoptotic death pathways [51], does not occur in GAEL-treated cells [51]. Thus, autophagy or altered lysosomal permeability do not appear to contribute to GAEL-induced EOC cell death.

A key requirement for GAEL cell killing activity is active endocytosis [98]. Inhibition of endocytosis with methyl-β-cyclodextrin completely blocked GAEL-induced vesicle formation and cell death. Additionally, experiments using temperature to block early endosome to late endosome maturation resulted in GAEL-induced vesicle formation but not cell death. While these observations intimately link endocytosis to GAEL activity, they do not reveal whether the requirement is just to transport GAELs into the cell or whether the requirement for endocytosis is related to their mechanism of cell killing or both. Ongoing studies utilizing proteomic and lipidomic analyses seek to identify specific pathways altered by *L*-Rham treatment to produce cell-killing effects in EOC cells. Preliminary evidence indicates that neither endoplasmic reticulum stress nor ferroptosis is involved in *L*-Rham-induced cell death. Collectively, these characteristics of GAEL-induced cell death led us to ask whether there is a known process that encompass these features.

The GAEL-induced phenotypic response is typical of a form of non-apoptotic, caspase-independent cell death called methuosis (Greek: *methuo*—to drink to intoxication) [99]. Numerous molecules have been identified that can induce methuosis including oncogenic *H-RAS* (Ras^G12V^) in glioblastoma cells [100], indole-based chalcones [101,102,103,104], methamphetamine [105], an ursolic acid derivative [106], the sphingolipid jaspine B [107], and miR-199a-3p [108]. These molecules induce LAMP1- and Rab7-positive, EEA1- and Rab5-negative cytosolic vesicle accumulation followed by caspase-independent cell death. It is proposed that cell death occurs because of the loss of endocytic vacuole maturation and recycling, which leads to metabolic failure [99]. Given that GAELs induce phase-lucent acidic vacuole formation and caspase-independent cell death similar to these agents, we suggest that GAELs may now be added to the list of methuosis-inducing agents.

While methuosis has been most extensively investigated in glioblastoma cell lines in vitro [100,103,104], additional cell models of methuosis have been tested in MDA-MB-231 and MCF-7 breast cancer cells [102], HCT116 colon cancer cells [102], HGC-27 gastric cancer cells [107], and papillary thyroid cancer-derived thyroid cells [108]. Importantly, methuosis inducers have been demonstrated to have significant antitumor properties in vivo against glioblastomas and breast cancer xenografts [102,103]. While *L*-Rham is capable of reducing tumor and ascites formation in vivo, further studies are required to determine if GAELs induce methuosis in target tumors in vivo.

Signaling molecules that contribute to methuotic cell death involve RAS and RAC signaling (critical for endocytosis) [100,105], c-Jun N-terminal kinase (JNK) [103], and PIK_fyve_ [101,103]. Inhibition of RAS signaling with farnesyl thiosalicylic acid or RAC signaling using EHT1864 or endocytosis using compounds such as Bafilomycin A_1_ abrogates vesicle formation and subsequent death induced by indole-based chalcones or methamphetamine [101,102,103,104,105]. The indole-based chalcone MOMIPP has been shown to induce increases in JNK1/2 phosphorylation and the downstream targets of JNK, including c-Jun and MKK4 [103]. Blocking JNK signaling with SP600125 protects cells from MOMIPP-induced cell death. PIK_fyve_ is a class III phosphoinositide kinase with activities that include conversion of phosphatidylinositol 3-phosphate to phosphatidylinositol 3,5-bisphosphate to regulate endosome trafficking. Inhibition of PIK_fyve_ activity pharmacologically using YM201636 [103] or genetically by CRISPR-based gene deletion [101] results in phase-lucent vacuole formation. Pharmacologic inhibition of PIK_fyve_ activity is also correlated with an increase in cell death [103]. Currently, we are examining whether JNK signaling and PIK_fyve_ activity contribute to *L*-Rham-induced cell death in EOC cells.

## 6. Conclusions: Utility of GAELs and Methuosis-Inducing Compounds for EOC Treatment

Once we identify the GAEL mechanism of action, this will open up the field to further develop agents that more specifically target GAEL effector pathways. Thus, additional in vitro and in vivo studies are required to understand how GAELs produce their cytotoxic effects and determine why GAELs are effective at killing platinum-resistant EOC cells. In addition, in vivo combination chemotherapy studies to determine whether GAELs may enhance the cytotoxic activity of standard treatment (carboplatin, paclitaxel) are warranted.

Our previous work has focused primarily on HGSOC. In order to broaden the applicability of *L*-Rham to other EOC histotypes, we tested whether *L*-Rham would induce cytotoxicity for all major histotypes of EOC. We determined that *L*-Rham effectively reduces cell viability in all histotypes, including poorly responsive mucinous and clear cell samples. (Figure 7). To quote insight from the Maltese research group, “[methuosis-inducing compounds] might serve as a prototype for new drugs that could be used to induce non-apoptotic death in cancers that have become refractory to agents that work through DNA damage and apoptotic mechanisms” [104]. Our finding that GAELs, and specifically *L*-Rham, can kill EOC xenografts in vivo and all histotypes of EOC in vitro lead us to advocate for the further investigation of methuosis-inducing agents by the EOC research community. We suggest that methuosis may be exploited as a novel treatment tactic for poorly responsive EOC histotypes as well as chemotherapy-resistant EOCs that will contribute to better outcomes for women experiencing EOC.

## Figures and Tables

**Figure 1 cancers-14-03318-f001:**
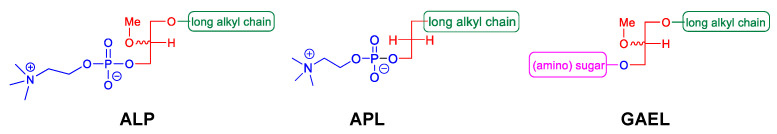
Different classes of antitumor ether lipids (AELs); alkyllysophospholipids (ALP), alkylphospholipids (APL), and glycosylated antitumor ether lipids (GAEL).

**Figure 2 cancers-14-03318-f002:**
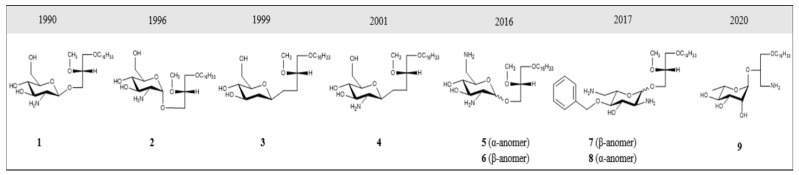
Structural evolution of GAEL prototypes. **1** = α-GLN, **2** = β-GLN; **3** and **4** are C-glycosidic analogs; **5** and **6** incorporate a second amino group at the 6 position; dibasic analogs **7** and **8** with *L*-glucose; **9** = 3-amino-1-O-hexadecyloxy-2R-(O–α-*L*-Rhamnopyranosyl)-sn-glycerol (*L*-Rham).

**Figure 3 cancers-14-03318-f003:**
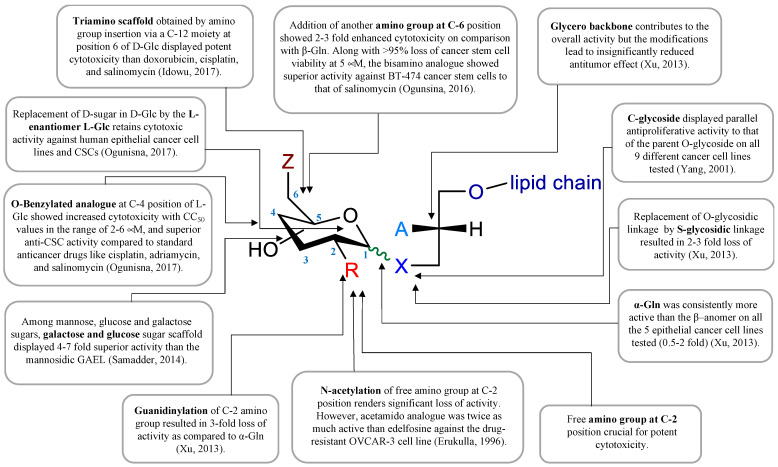
GAEL structure–activity relationship map. Data compiled from references [50,52,88,90,91,92,94].

**Figure 4 cancers-14-03318-f004:**
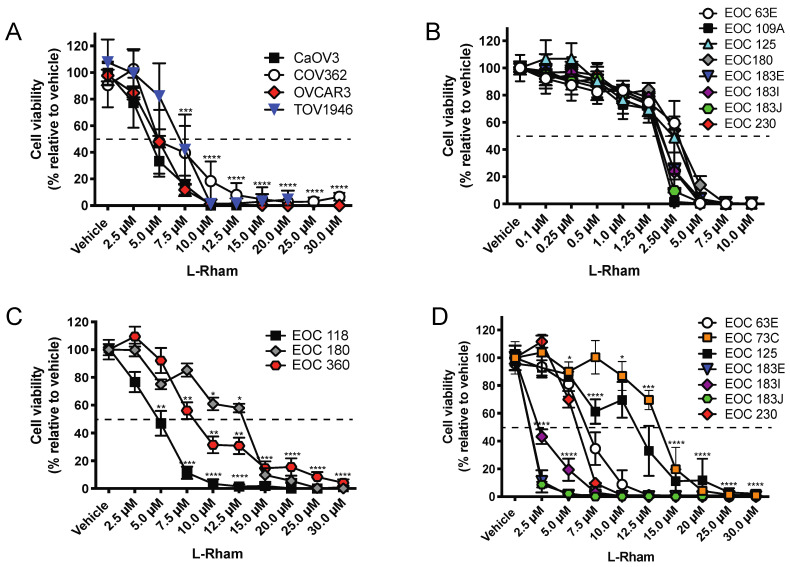
HGSOC cell viability after *L*-Rham treatment. (**A**). Dose response of established HGSOC cell lines treated with *L*-Rham for 48 h. Cells were grown as adherent monolayers. For (**A**,**B**), cells were grown as adherent monolayers, and for (**C**,**D**) cells, they were grown as non-adherent cell aggregates or spheres (depending on the individual sample). The dashed lines indicate 50% cell viability. Asterisks (*) indicate significant differences from vehicle-treated cells [* < 0.05, ** < 0.01, *** < 0.001, **** < 0.0001]; 2-way ANOVA with multiple comparisons was used to assess significance. (**B**). Dose response of chemosensitive and chemoresistant HGSOC patient samples treated with *L*-Rham for 48 h. (**C**). Dose response of chemosensitive HGSOC patient samples treated with *L*-Rham for 48 h. (**D**). Dose response of chemoresistant HGSOC patient samples treated with *L*-Rham for 48 h. Modified from reference [49], with permission.

**Figure 5 cancers-14-03318-f005:**
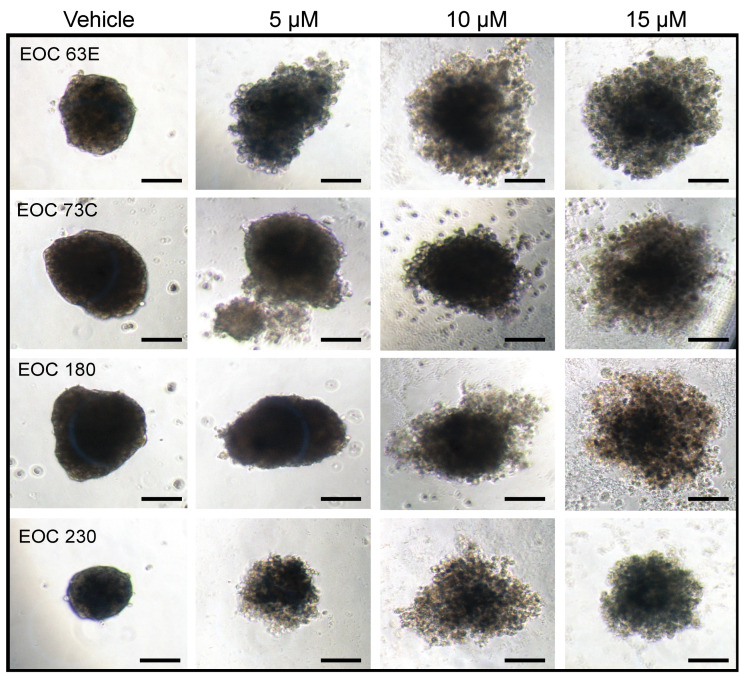
HGSOC cell viability after *L*-Rham treatment. Alteration in spheroid morphology in response to increasing doses of *L*-Rham after 48 h. Solid bar = 100 µm. Modified from reference [49], with permission.

**Figure 6 cancers-14-03318-f006:**
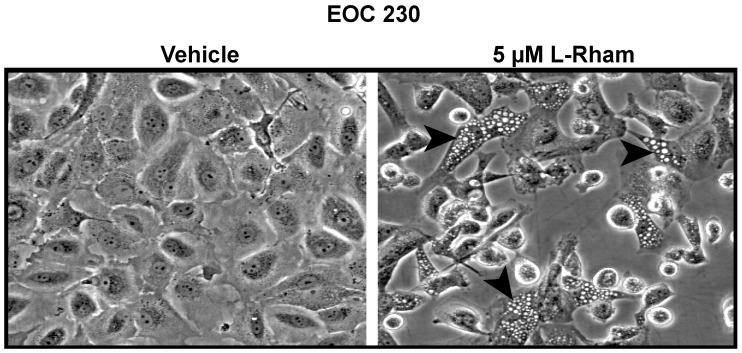
*L*-Rham phenotypic response. Development of phase-lucent cytoplasmic vesicles (arrowheads) 24 h after initial exposure to *L*-Rham. Primary EOC230 cells were isolated from HGSOC patient ascites. Vehicle = 0.9% saline. Solid bar = 100 µm.

**Figure 7 cancers-14-03318-f007:**
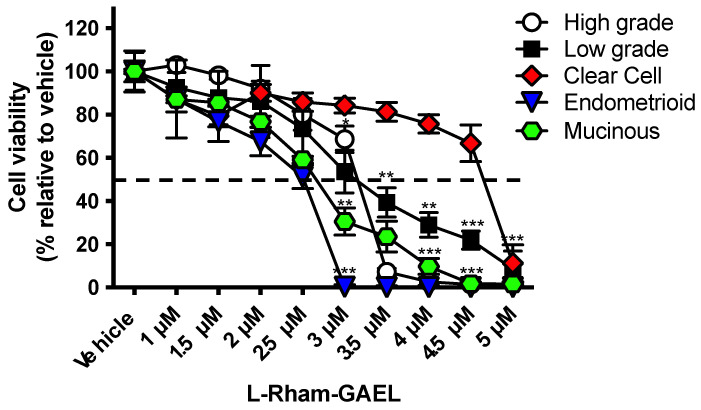
Primary patient EOC cell *L*-Rham sensitivity. Dose response to *L*-Rham for primary patient samples isolated from the ascites of patients representing each major EOC histotype. Vehicle = 0.9% saline. Cells were grown as a 2D adherent monolayer and exposed to *L*-Rham for 48 hours. The dashed lines indicate 50% cell viability. Data were obtained from 3 independent experiments with 6 data points per experiment. Vertical lines indicate standard deviation, and asterisks (*) indicate significant difference from vehicle-treated cells, [* < 0.05, ** < 0.01, *** < 0.001]; Kruskal–Wallis test with Dunn’s multiple comparisons per group was used to assess significance.

**Table 1 cancers-14-03318-t001:** Summary of *L*-Rham efficacy on OVCAR3 i.p. xenograft in NRG mice.

Group	Group Name	Dose Schedule/Termination	Mice with Solid Tumors	Ascites (%)
2 *	Vehicle—2	Q4Dx8 Termination: day 74 ***	100% (12/12) ^@^	92% (11/12)
4 *	*L*-Rham—low 2	Q4Dx8 Termination: day 74 ***	33% (2/6)	0% (0/6)
5 **	*L*-Rham—high	Q4Dx8 Termination: day 74 ***	100% (6/6)	0% (0/6)

* treatment began 7 days post OVCAR-3 cell inoculation i.p. = low tumor burden = “low”. ** treatment began 40 days post OVCAR-3 cell inoculation i.p. = high tumor burden = “high”. *** termination of mice was 6 days post final drug administration. ^@^ numbers in parentheses reflect number of mice.

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
