# Peer review of "The Potential of Novel Lipid Agents for the Treatment of Chemotherapy-Resistant Human Epithelial Ovarian Cancer"

_cancers, 2022, doi:10.3390/cancers14143318_

Round 1

Reviewer 1 Report

Mark and colleagues introduced the role of GAELs in chemo-therapy-resistant human epithelial ovarian cancer, and the overall description of the article is clear. However, the current version is inappropriate as a review, it has the following main problems that confuse me.

1. I have no doubts about the introduction and parts 2-3 of this article. In part 4, the authors adopted a series of experimental evidence to prove the functions of GAELs on EOC, but there is a lack of sufficient research on the mechanism of drug action in EOC. To illustrate the mechanism of action of GAELs on EOC, the authors cited some research showing that GAELs act on other cancer cells (Doi:10.1039/c6md00328a. Doi: 10.1021/acs.jmedchem.6b01773. Doi: 10.1016/j.ejmech.2014.03.057. Doi: 10.4161/auto.9120). The mechanism of action is based on the content published by the author in 2017 (Doi: 10.1186/s13046-017-0538-9). To explain the mechanism of GAELs on EOC, the authors simply repeated the conjecture of the effect of GAELs on other tumors and I didn't get much more from this article.

2. This article lacks clinical trial data and related research, pharmacokinetics and pharmacodynamics of GAELs are also not well described.

3. This article lacks summary and comparison of the current treatment of EOC. Overall, the authors seem to be more focused on introducing the effects of drugs on EOC via their existing research content, and the current version seems to be more suitable for resubmission in other article formats rather than as a review.

Author Response

Reviewer #1

Mark [Nachtigal] and colleagues introduced the role of GAELs in chemo-therapy-resistant human epithelial ovarian cancer, and the overall description of the article is clear. However, the current version is inappropriate as a review, it has the following main problems that confuse me.

  1. I have no doubts about the introduction and parts 2-3 of this article. In part 4, the authors adopted a series of experimental evidence to prove the functions of GAELs on EOC, but there is a lack of sufficient research on the mechanism of drug action in EOC. To illustrate the mechanism of action of GAELs on EOC, the authors cited some research showing that GAELs act on other cancer cells (Doi:10.1039/c6md00328a. Doi: 10.1021/acs.jmedchem.6b01773. Doi: 10.1016/j.ejmech.2014.03.057. Doi: 10.4161/auto.9120). The mechanism of action is based on the content published by the author in 2017 (Doi: 10.1186/s13046-017-0538-9). To explain the mechanism of GAELs on EOC, the authors simply repeated the conjecture of the effect of GAELs on other tumors and I didn't get much more from this article.

We respect the Reviewer’s opinion. Since publishing the most recent paper [doi 10.1016/j.tranon.2021.101203], our work has been receiving a great deal of attention at meetings with basic, translational and clinical scientists, as well as receiving many invitations to present our work at international conferences. Therefore, many scientists appear to appreciate the research on GAELs and their potential as a putative area for further investigation to treat human epithelial ovarian cancer patients. Since the majority of the research on this topic has been produced in the author’s laboratories we cite this peer-reviewed research as evidence for the GAEL mechanism of action.

At this time the exact GAEL mechanism of action remains unknown. What we know about the GAEL mechanism of action is discussed in section 5 based on previously published peer-reviewed literature as is befitting for a review article. The putative mechanism of action for GAELs is based on our research over the past 17 years, and for L-Rham specifically was based on what we tested in 2017 and for the 2021 article. Further research has been conducted for L-Rham recently that rules out additional pathways or biological processes including ferroptosis, PI3K, AKT, and JNK signaling; however, this research has not been peer-reviewed (confidential) and will be included in a future publication submitted for peer-review.

  1. This article lacks clinical trial data and related research, pharmacokinetics and pharmacodynamics of GAELs are also not well described.

GAELs are still in the preclinical developmental stages. This article describes preclinical evidence, including in vivo xenograft models, indicating that GAELs are a potential novel drug class that may be developed for future clinical trials. No work on clinical trials has been done to date because of the need to obtain appropriate preclinical data (indicated by our own practice for drug development and after consultation with the pharmaceutical industry). Pharmacokinetic and pharmacodynamic studies are slated for future research.

  1. This article lacks summary and comparison of the current treatment of EOC. Overall, the authors seem to be more focused on introducing the effects of drugs on EOC via their existing research content, and the current version seems to be more suitable for resubmission in other article formats rather than as a review.

We were directed by the Guest Editor not to include a “summary and comparison of the current treatment of EOC” since other articles in this Special Issue of Cancers on New Insights of Ovarian Cancer Treatment would already cover that topic.

Reviewer 2 Report

I am happy to read your insightful work introducing GAEL, a new promising anticancer candidate for EOC. Here is my recommendation as follows;

Is there any possibility GAEL kills normal cells rather than cancer cells? I think prohibiting endocytosis, MOA of GAEL, can kill all live cells. How can you demonstrate the cancer-specific killing of GAEL?

Author Response

Reviewer #2

I am happy to read your insightful work introducing GAEL, a new promising anticancer candidate for EOC. Here is my recommendation as follows:

  1. Is there any possibility GAEL kills normal cells rather than cancer cells?

As with any potentially cytotoxic agent, including all cytotoxic agents used to treat EOC, there is a reasonable expectation that they will also cause toxicity to normal cells. We fully expect that at high doses, GAELs will kill normal cells; however, our in vivo maximum tolerated dose studies indicate that these drugs are well tolerated by the mice with no significant toxicity beyond a generalized peritonitis (detailed in reference 91). Tests to determine maximum tolerated dose are required for all novel pharmaceuticals. Our unpublished research and published research [doi 10.1016/j.tranon.2021.101203] indicate a high dosage tolerance in vivo for GAELs, indicating that any cell killing of normal cells is well tolerated by the animal (tested in Rag2M and NSG immune-compromised mice). Further evidence was also provided by our chicken allantoic membrane xenograft studies [included in reference 91 doi 10.1016/j.tranon.2021.101203] that “no toxicity or macroscopic effect on the development of chick embryos was observed.” Thus, we were able to identify a dose of L-Rham suitable to effect cancer cells that were well tolerated or had no effect on normal cells.

  1. I think prohibiting endocytosis, MOA of GAEL, can kill all live cells. How can you demonstrate the cancer-specific killing of GAEL?

We agree, inhibiting endocytosis would likely kill all cells. For accuracy, we believe the MOA of GAELs will most likely involve a dose-dependent disruption in the maturation and recycling of endocytic vesicles, as per the MOA for methuosis.

Please note that GAELs were not developed as targeted cancer agents, rather they would fall under the class of general cytotoxic agents, similar to carboplatin or paclitaxel. To ensure this point is clear in the article, we have added the following statement:

Lines 79-80, “GAELs would fall under the class of cytotoxic rather than targeted agents.”

Moreover, similar to our response to question #1, these drugs are well tolerated in mice and show “no toxicity or macroscopic effect on the development of chick embryos was observed.”

Reviewer 3 Report

This review article describes the capability of glycosylated antitumor ether lipids (GAELs) in the treatment of epithelial ovarian cancer resistant to chemotherapy. Authors have described the development of GAELs, structure activity relationship, the specific ability of them to act against EOC and provide therapeutic benefits with evidence from in vitro and in vivo experiments. The overall context is of  importance and is novel, specifically considering the anti-apoptotic mechanism of action and the effect of inducing methuosis. Authors have included the relevant literature and the manuscript structure is organized in a good manner. However, it is important to consider that there is a need for more number of studies to completely understand the mechanism and clinical feasibility. The manuscript can be considered a good attempt consolidating the existing studies with GAELs, however there are few concerns which can be addressed by the authors and they are given below:

1.     Simple summary section: “Importantly, we have shown that GAELs are effective at killing chemotherapy- resistant EOC cells in the lab and in a mouse avatar” – it is better to use the terms in vitro and in vivo for the scientific context.

2.     Graphical abstract can be further modified. Please include more details beneath the images (for example: changes in cell morphology can give a broader idea about mechanism from the abstract). Please indicated percentage figures with standard deviation for both tumors and ascites.

3.     Since introductory section focusses on EOC and also therapeutic approaches offering improvement in survival rates; it would be ideal to add few more statements which covers approaches such as intraperitoneal therapy with surgical debulking which has shown considerable improvements. This would also give a background on localized administration approaches which can also provide therapeutic advantages for EOC treatment. 

4.     Since there is strong data with GAELs with regard to tumor and ascites development, it would be good to briefly discuss the advantages with respect to the EOC stages. Please provide a background on this as well as metastasis and ascites development are largely confined to stages 3 and 4 of EOC.

5.     What would be the major advantages of relying on an apoptosis-independent mechanism? The effects have been explained in the subsequent sections; however it is important to briefly explain this in the introductory section as well for the readers to understand the primary benefits of GAELs in comparison to conventional drugs working with apoptotic mechanisms.

6.     ‘Methuosis’ needs to be better explained in simple terms- there are few sections like page 10, paragraph 2- however this does not provide a complete picture.

7.     Usage of tables is highly recommended- so as to provide details in a compact and concise manner. Please consider restructuring section 2 into a table; this would increase the readability of the article as well.

8.     What could have been the rationale for choosing 8 days of treatment in the experiment with eggs (page 8, paragraph 1)? 

9.     There should be more discussion regarding the possibility of toxicity with GAEL treatment. Page 8, paragraph 2 talks about mild clinical signs of toxicity. Has there been any further reports in this direction? Was this specifically assessed in in vivo experiments? 

10.  Table 1 given from reference 90 can be given as separate table and not as an image. 

11.  Table 1: how was ascites% determined, please provide standard deviation as applicable.

12.  Page 8, paragraph 2: ‘Drug delivery (intravenous or intraperitoneal) and drug vehicle (1% propylene glycol or 0.9% saline) studies were conducted’ – what were the drug doses used for intravenous and intraperitoneal administrations?

13.  Would administration routes (intravenous vs intraperitoneal) change the toxicity profiles of GAELs? Has this been assessed in any previous studies?

14.  Figures 4 and 7: statistical analyses used for the data should be indicated in legends. Please include the reference for figure 7.

15. It would be better to include a conclusive paragraph describing the gaps in the current studies and indicate the need for more preclinical and clinical studies to completely understand the mechanistic pathways and clinical feasibility of utilizing GAELs for EOC treatment. 

Author Response

Reviewer #3

This review article describes the capability of glycosylated antitumor ether lipids (GAELs) in the treatment of epithelial ovarian cancer resistant to chemotherapy. Authors have described the development of GAELs, structure activity relationship, the specific ability of them to act against EOC and provide therapeutic benefits with evidence from in vitro and in vivo experiments. The overall context is of importance and is novel, specifically considering the anti-apoptotic mechanism of action and the effect of inducing methuosis. Authors have included the relevant literature and the manuscript structure is organized in a good manner. However, it is important to consider that there is a need for more number of studies to completely understand the mechanism and clinical feasibility. The manuscript can be considered a good attempt consolidating the existing studies with GAELs, however there are few concerns which can be addressed by the authors and they are given below:

  1. Simple summary section: “Importantly, we have shown that GAELs are effective at killing chemotherapy- resistant EOC cells in the lab and in a mouse avatar” – it is better to use the terms in vitro and in vivo for the scientific context.

Thank you. This has been changed.

2.Graphical abstract can be further modified. Please include more details beneath the images (for example: changes in cell morphology can give a broader idea about mechanism from the abstract). Please indicated percentage figures with standard deviation for both tumors and ascites.

Our understanding is that a graphical abstract provides limited written details. Changes in cell morphology are clear from the images; however, we have added a question mark (“?”) to the term “Methuosis” under the images showing changes in cell morphology. We hope that this will peak the reader’s interest to read further.  Details regarding this cell response are provided in the body of the text.

The percentage figures reflect absolute values.  Further details are provided in the response to question # 11.

  1. Since introductory section focusses on EOC and also therapeutic approaches offering improvement in survival rates; it would be ideal to add few more statements which covers approaches such as intraperitoneal therapy with surgical debulking which has shown considerable improvements. This would also give a background on localized administration approaches which can also provide therapeutic advantages for EOC treatment. 

We were directed by the Guest Editor not to include a “summary and comparison of the current treatment of EOC” since other articles in this Special Issue on New Insights of Ovarian Cancer Treatment would already cover that topic.

  1. Since there is strong data with GAELs with regard to tumor and ascites development, it would be good to briefly discuss the advantages with respect to the EOC stages. Please provide a background on this as well as metastasis and ascites development are largely confined to stages 3 and 4 of EOC.

Given that we observed the best in vivo response to L-Rham treatment in the low tumour burden model, we had proposed that GAEL treatment might be best as an adjuvant therapy following optimal surgical debulking and initial rounds of standard chemotherapy (Carboplatin + paclitaxel). Lines 312-317 “Our research suggests that L-Rham might best serve as an adjuvant agent following optimal surgical debulking and standard-of-care chemotherapy (carboplatin + paclitaxel) in order to produce a low tumour environment where L-Rham would effectively reduce tumour recurrence and block ascites formation.”

Thus, Stage would not have an impact if there was optimal cytoreduction.

  1. What would be the major advantages of relying on an apoptosis-independent mechanism? The effects have been explained in the subsequent sections; however it is important to briefly explain this in the introductory section as well for the readers to understand the primary benefits of GAELs in comparison to conventional drugs working with apoptotic mechanisms.

We have added the following statement to indicate the advantage of using a non-apoptoic cell death mechanisms for cancer treatment.  Lines 85-89, “Because cancer cells often overcome apoptosis induced by many cytotoxic agents, addition of GAELs would be an excellent complementary treatment to overcome drug-resistance mechanisms. Moreover, as cellular genetic heterogeneity is observed in EOCs, the ability of GAELs to kill chemotherapy-sensitive and resistant cells would be an additional benefit for including GAELs as adjuvant treatment.”

  1. ‘Methuosis’ needs to be better explained in simple terms- there are few sections like page 10, paragraph 2- however this does not provide a complete picture.

We have described the most up-to-date and complete information on methuosis and the cellular responses leading to methuosis, and included historical and recent references for this process. We have also added the following statement to add clarity:

Lines 369-370 “It is proposed that cell death occurs because of the loss of endocytic vacuole maturation and recycling that leads to metabolic failure (101).”

  1. Usage of tables is highly recommended- so as to provide details in a compact and concise manner. Please consider restructuring section 2 into a table; this would increase the readability of the article as well.

It is unclear how we would restructure section 2 into a Table and believe that the text describing the history of antitumor ether lipid evolution provides the reader with a more comprehensive description of these historical events.

  1. What could have been the rationale for choosing 8 days of treatment in the experiment with eggs (page 8, paragraph 1)?

The rationale is provided in the article where the COV362 CAM xenograft was published in reference 91 [doi 10.1016/j.tranon.2021.101203]. In brief, the xenograft was initiated at day 9 of chick development, drug treatment was started on day 11 and eggs were treated every two days for 4 treatments total. Eggs were harvested at day 18 prior to full chick development (and hatching) at day 21, which complies with European Directive 2010/63/EU on the use of animals for research.

For clarity, we have added the amended the previous statement to now read Lines 277-278, “Eggs were treated for a total of 8 days, and harvested at day 18 prior to normal hatching at day 21.”

  1. There should be more discussion regarding the possibility of toxicity with GAEL treatment. Page 8, paragraph 2 talks about mild clinical signs of toxicity. Has there been any further reports in this direction? Was this specifically assessed in in vivo experiments? 

Drug tolerability and toxicity in vivo is extensively discussed in reference 91 [doi 10.1016/j.tranon.2021.101203]. Supplementary data from reference 91, Table S5 and Table S8, outline the specific in vivo responses to drug treatment for each mouse.

  1. Table 1 given from reference 90 can be given as separate table and not as an image. 

We have recreated this as a Table in the Text.

  1. Table 1: how was ascites% determined, please provide standard deviation as applicable.

The number of mice that had ascites or solid tumors was identified at necropsy.  The percent numbers provided are absolute values (e.g., 92% = 11/12 animals) and so there is no standard deviation. This data is provided for each animal in Table S7 of reference 91 [doi 10.1016/j.tranon.2021.101203]. We have amended Table 1 to reflect the absolute number in parentheses beside the percent values.

  1. Page 8, paragraph 2: ‘Drug delivery (intravenous or intraperitoneal) and drug vehicle (1% propylene glycol or 0.9% saline) studies were conducted’ – what were the drug doses used for intravenous and intraperitoneal administrations?

This data is provided for each animal in Table S5 of reference 91 [doi 10.1016/j.tranon.2021.101203]. For clarity we have added the reference to this statement.

  1. Would administration routes (intravenous vs intraperitoneal) change the toxicity profiles of GAELs? Has this been assessed in any previous studies?

This was not assessed in previous studies. Reference 91 is the first study showing GAEL tolerability in vivo, which was why we published the different vehicles, delivery methods and dosages.

  1. Figures 4 and 7: statistical analyses used for the data should be indicated in legends. Please include the reference for figure 7.

The statistical methods have been added to the figure legends.

Figure 7 is an original figure generated for this Review article and so there is no previous reference.

  1. It would be better to include a conclusive paragraph describing the gaps in the current studies and indicate the need for more preclinical and clinical studies to completely understand the mechanistic pathways and clinical feasibility of utilizing GAELs for EOC treatment.

Good idea, thank you.  We have added the following to Section 6: Lines 402-410, “Once we identify the GAEL mechanism of action this will open up the field to further develop agents that more specifically target GAEL effector pathways. Thus, additional in vitro and in vivo studies are required to understand how GAELs produce their cytotoxic effects and determine why GAELs are effective at killing platinum-resistant EOC cells. In addition, in vivo combination chemotherapy studies to determine whether GAELs may enhance the cytotoxic activity of standard treatment (carboplatin, paclitaxel) are warranted.”

Reviewer 4 Report

This is a very well written paper presenting evidence of a lipid agent that has been shown to be effective against chemoresistant EOC. The write up is very comprehensive and the evidence are well presented.

Couple of minor points:

1. It would be good if the authors could discuss the specificity and selectivity of this agent in terms of targeting. How do these agents only specifically target chemoresistant cells (and/or cancer like stem cells) but not healthy cells?

2. Once administered, do these agents cross the blood brain barrier? Is there any evidence? A brief discussion would be useful.

3. Figure 3- top left corner- the box is not fully formed- just needs a minor editing here.

Author Response

Reviewer #4

This is a very well written paper presenting evidence of a lipid agent that has been shown to be effective against chemoresistant EOC. The write up is very comprehensive and the evidence are well presented.

 Couple of minor points:

  1. It would be good if the authors could discuss the specificity and selectivity of this agent in terms of targeting. How do these agents only specifically target chemoresistant cells (and/or cancer like stem cells) but not healthy cells?

Please note that GAELs were not developed as targeted cancer agents, rather they would fall under the class of general cytotoxic agents, similar to carboplatin or paclitaxel. To ensure this point is clear in the article, we have added the following statement:

Lines 79-80, “These compounds would fall under the class of cytotoxic agents rather than targeted agents.”

As with any potentially cytotoxic agent, including all cytotoxic agents used to treat EOC, there is a reasonable expectation that they will also cause toxicity to normal cells. Tests to determine maximum tolerated dose are required for all novel pharmaceuticals. Unpublished research and published research [doi 10.1016/j.tranon.2021.101203] indicate a high dosage tolerance in vivo for GAELs, indicating that any cell killing of normal cells is well tolerated by the animal.  Further evidence was also provided by our chicken allantoic membrane xenograft studies [included in doi 10.1016/j.tranon.2021.101203] that “no toxicity or macroscopic effect on the development of chick embryos was observed.” Thus, we were able to identify a dose of L-Rham suitable to effect cancer cells that were well tolerated or had no effect on normal cells.

  1. Once administered, do these agents cross the blood brain barrier? Is there any evidence? A brief discussion would be useful.

No previous or subsequent studies have been done to determine if these drugs cross the blood brain barrier. Rather, the in vivo work that has been conducted focused on affects of tumors in the peritoneal cavity, the primary site for EOC tumor development and metastasis. We hope that this article will inspire additional researchers to test GAELs as novel treatments for their cancers, including primary or metastatic cancers in the brain.

  1. Figure 3- top left corner- the box is not fully formed- just needs a minor editing here.

Thank you, this has been corrected.

Round 2

Reviewer 1 Report

 I have no concerns about its publication

Reviewer 3 Report

The authors have taken good efforts to answer all questions raised in the previous review with adequate explanations. Relevant statements have been added to the manuscript with adequate literature support. The manuscript in its current form can be considered for possible acceptance.